# A Virtual Sensor for Collision Detection and Distinction with Conventional Industrial Robots

**DOI:** 10.3390/s19102368

**Published:** 2019-05-23

**Authors:** Zhijing Li, Jinhua Ye, Haibin Wu

**Affiliations:** School of Mechanical Engineering and Automation, Fuzhou University, Fuzhou 350116, China; lizhijingwei@163.com (Z.L.); yejinhua@fzu.edu.cn (J.Y.)

**Keywords:** virtual sensor, observers, collision detection and distinction, industrial robots

## Abstract

Physical contact inevitably occurs during robot interaction with outside environments. A robot should have the ability to detect and distinguish whether a physical interaction between a human and the robot is contact or collision, so as to ensure human safety and improve interaction performance. In this paper, a virtual sensor that can detect and distinguish contact and collision between humans and industrial robots is proposed. Based on the generalized momentum of the robot, two observers with low-pass and band-pass filter characteristics were designed in this virtual sensor to realize the robot collision detection. Using the different frequency distribution ranges of the lighter contact force signal and the heavier collision force signal, the filter parameters in the two observers were appropriately selected to distinguish between collisions and contacts in close interaction between humans and robots. The virtual sensor does not require acceleration information or inverse dynamics calculations. It only needs to sample the motor driving current and position information of the robot joint, and can easily be applied to conventional industrial robots. The experimental results show that the low-pass and band-pass torque observers can detect different force signals in real-time, and the proposed virtual sensor can be used for collision detection and distinction in human–robot interactions.

## 1. Introduction

With the rapid development of robotics, the application of robots has expanded from traditional industrial manufacturing to public services, entertainment, and medical care fields. Whether they are employed in production or service, the interaction and cooperation between humans and robots have become an important direction for the development of current and future robotics [1]. In the process of human–robot interaction sharing a limited physical space, the safety of the robot, and especially human safety, should be ensured first to prevent the robot from accidentally colliding with humans or the surrounding environment during the movement. At the same time, intentional contact between humans and robots is necessary to realize the transmission of information between them, such as robots teaching through direct drag. This is the premise and basis of human–robot interaction and Cobot technology [2].

Most of the existing research focuses on how to prevent robots from colliding with humans or the surrounding environment during work, or how to reduce the collision force in the case of collision [3,4,5,6]. Many studies have adopted new cooperative robots to improve the safety of human–robot interaction; examples include the KUKA LWR robot and Rethink robot [7]. This type of cooperative robot integrates several advanced force or torque sensors, providing many advantages for robot collision detection and safety response [6]. At present, although cooperative robots have made great progress, their prices are too high for many applications, so they are still in the stage of research or small-scale experiments. Conventional industrial robots are used in most applications, which typically do not include force or torque sensors in order to keep costs to a minimum. A common method of robot collision detection is to install a multidimensional torque sensor at the end of the robot. The disadvantage of this method is that force sensors are expensive and can only detect collisions at the end of the robot. On the other hand, collision detection can also be achieved through indirect information that is closely related to the joint torque. Je et al. studied a robot collision detection method based on motor current variation [8]. The robot collision methods proposed in Refs. [9,10] are also based on the method of measuring the joint current. The model-based approach is relatively simple compared to methods of analyzing current signals. Therefore, more model-based methods are used in robot collision detection [11,12]. Usually, a model-based method requires joint acceleration information, which introduces large calculation errors (the acceleration is normally derived from the second derivative of the displacement sampled from the encoder). To this end, the generalized momentum-based method can be used to avoid acceleration calculation [13,14]. 

Although the above approaches can play a vital role in improving the safety of robots, they cannot achieve intentional contact and information interaction between humans and robots. In fact, in a human–robot interaction environment, workers often need to directly drag and guide the robot to perform specific actions, including the task of carrying or installing workpieces [15]. These work scenarios require intentional contact between robots and humans or the unknown environment. Geravand et al. proposed a good method to distinguish contact and collision between humans and robots through directly filtering the residual current of the robot joint motor [16]. Indriet al. further improved this method and proposed a sensor-less methodology implementing virtual sensors to manage both collision detection and manual guidance sessions [17]. Kouris et al. proposed a force signal analysis method to improve the real-time performance of distinguishing between contact and collision [18]. As mentioned in Ref. [18], the proposed method requires a higher computational performance of the control system. Golz et al. also used mechanical learning to achieve contact and collision discrimination [19], but it requires a contact model and prior training. Moreover, there are few studies on improving the safety interaction performance of conventional industrial robots. Therefore, further research is needed.

In this paper, contact is defined as a force between a person and a robot that has a certain amplitude and a relatively long duration. Collision is defined as a force between a robot and a person that has a large amplitude and a short duration. Contact and collision cannot be effectively distinguished by simply setting a threshold. The main difference between contact and collision is that the force signals of the two have different distribution characteristics in the time domain and the frequency domain. The contact force changes relatively slowly, and the collision force has a large mutation. Inspired by Ref. [16], a new virtual sensor for collision detection and distinction is proposed. Two generalized momentum-based torque observers with low-pass and band-pass filtering characteristics are contained in this virtual sensor. It does not require force sensors, does not require acceleration information, and does not need inverse dynamics calculations. It can be implemented on the widely used conventional industrial robot (it only needs to be equipped with axis encoders and servomotors). Compared with the existing methods, the proposed virtual sensor can not only detect a collision between a human and a robot in real-time to ensure the safety of the human body, but it can also distinguish contact from collision and improve the human–robot interaction performance. 

The rest of the paper is organized as follows. In Section 2, a virtual sensor for collision detection and distinction is proposed and the specific design and implementation process of the virtual sensor is described in detail. The robot joint friction identification and compensation methods are presented in Section 3 to improve the performance of the proposed virtual sensor. The effectiveness of the virtual sensor was verified by several experiments, and the corresponding experimental results are presented in Section 4. The concluding comments of the paper are provided in Section 5. 

## 2. The Virtual Sensor Development

### 2.1. Research Ideas of the Virtual Sensor for Collision Detection and Distinction

According to the results of Refs. [16,17,18], the noise signals in a robot system are concentrated in the high-frequency region. The intentional human contact force signals are mainly distributed in the low-frequency region. The frequency component of the collision force signals is higher than the intentional contact force signal. Therefore, a virtual sensor containing low-pass and band-pass filtered observation characteristics can be used to observe the force signals with different frequency distribution regions. The specific design scheme of the virtual sensor is shown in Figure 1. 

Figure 1 depicts the observations of the proposed virtual sensor for external force signals at different frequencies. It can be seen from Figure 1 that the low-pass filter observer can observe whether there is contact force or collision force between the human and the robot, but it is impossible to distinguish between the two. The band-pass filter observer filters out low-frequency force signals and detects high-frequency components of the accidental collision force signal. Therefore, the use of these two observers in the virtual sensor can achieve the detection and distinction of intentional contact and accidental collision between humans and robots.

### 2.2. Robot Dynamics Modeling

When a contact or collision occurs between the robot and the unknown environment, the dynamic equation of the *n* degrees of freedom (DOF) robot based on the Lagrangian method can be written as
(1)$$D(q)\ddot{q}+C(q,\dot{q})\dot{q}+g(q)+\tau_{f}=\tau_{m}-\tau_{c}\left(\right),$$
where q, $\dot{q}$, and $\ddot{q}\in R^{n}$ are the position, velocity, and acceleration of the robot joint, respectively, $D(q)\in R^{n\times n}$ is a symmetric positive definite inertia matrix, $C(q,\dot{q})\dot{q}\in R^{n\times n}$ is the Coriolis and centrifugal forces term, $g(q)\in R^{n}$ is the gravity term, $\tau_{f}\in R^{n}$ is the frictional term, $\tau_{m}\in R^{n}$ is the driving torque of the joint, and $\tau_{c}\in R^{n}$ is the equivalent torque generated by the external contact or collision force F in the corresponding joint of the robot, and its expression is
(2)$$\tau_{c}=J^{T}F=[\tau_{c}{}_{1}\cdots \tau_{c}{}_{i},0\cdots 0],$$
where $J^{T}$ is the transposed Jacobian matrix.

According to Ref. [3], $\dot{D}(q)-2C(q,\dot{q})$ is the skew-symmetric matrix, so its equivalent expression is given by
(3)$$\dot{D}(q)=C(q,\dot{q})+C^{T}(q,\dot{q}).$$

Since conventional industrial robots generally do not have a joint torque sensor installed, the joints are usually composed of a servomotor, a transmission mechanism, and an output link. In order to obtain the joint control torque in the dynamics equation (Equation (1)), a method of measuring the joint motor current is employed. The driving current of the motor is converted accordingly, and the equation for calculating the joint driving torque is
(4)$$\tau_{m}=n_{har}T_{m}I_{m},$$
where $I_{m}$ is the current of the motor, $T_{m}$ is the correlation coefficient of the current into the torque, and $n_{har}$ is the transmission ratio of the joint motor to the output link. 

### 2.3. Low-Pass Torque Observer Design

As shown in Figure 1, the proposed virtual sensor requires an observer to detect the equivalent torque $\tau_{c}$ generated on the joint when a contact or collision occurs. A torque observer with low-pass filtering characteristics was designed based on the generalized momentum. According to Refs. [13,14], the generalized momentum in the robot dynamics system is defined as
(5)$$P=D(q)\dot{q}.$$

The time derivative of Equation (5) yields
(6)$$\dot{P}=\dot{D}(q)\dot{q}+D(q)\ddot{q}.$$

Using Equations (1), (3), and (6), one obtains
(7)$$\left\{ \begin{array}{l} \dot{P}=n_{har}T_{m}I_{m}-\tau_{c}-h(q,\dot{q})\\ h_{i}(q_{i},{\dot{q}}_{i})=-C_{i}^{T}(q_{i},{\dot{q}}_{i}){\dot{q}}_{i}+\tau_{f}{}_{i}+g_{i}(q_{i}) \end{array}\right.,$$
where i=1⋯n. When the robot is subjected to an external force, its real-time momentum will cause a sudden change. Therefore, the external momentum deviation observation vector of the robot can be defined as
(8)$$r=K(\hat{P}-P),$$
where K is a positive definite diagonal gain matrix and $\hat{P}$ is the estimated value of the robot generalized momentum. Using Equations (7) and (8), and replacing r with $\tau_{c}$, we obtain
(9)$$\dot{\hat{P}}=\tau -h(q,\dot{q})-r.$$
Then, Equation (8) can be rewritten as
(10)$$r=K(\int (\tau -h(q,\dot{q})-r)dt-P).$$

The working principle diagram of the momentum deviation observer is shown in Figure 2.

From Equation (10), the differential calculation for r can be obtained as
(11)$$\dot{r}=-Kr+K\tau_{c}.$$

Performing a Laplace transform on the above equation, the transfer function of the observer can be obtained as
(12)$$\frac{r_{i}(s)}{\tau_{ci}(s)}=\frac{K_{i}}{s+K_{i}},i=1,2,\cdots n,$$
where $K_{i}$ is the i row element in the gain matrix K. As can be seen from the above equation, the observer is a first-order low-pass filter system, and its observation corresponds to the torque $\tau_{c}$. Therefore, this observer is referred to herein as a low-pass torque observer (LPTOB). Appropriately taking the value of $K_{i}$, the observer can not only detect a collision between the human and the robot in real-time, but also can filter out the noise signal, whose frequency is higher than $K_{i}$ in the control system. The following is a specific collision judgment method. The observations of the LPTOB can be written as
(13)$$r=\left[r_{1}\cdots ,r_{i},r_{i+1},\cdots r_{n}\right],$$
where i=1,2,⋯n. When the i link is involved in a collision, comparing the elements in r from $r_{n}$ to $r_{1}$ one by one, we have
(14)$$\left|r_{i}\right|>N_{i}\;\mathrm{and}\;\left|r_{i+1}\right|<N_{i+1},i=1,2,\cdots n,$$
where $N_{i}$ is the joint collision detection threshold. Then, the corresponding collision direction can be determined by the positive or negative value of $\frac{r_{i}}{\left|r_{i}\right|}$.

### 2.4. Band-Pass Torque Observer Design

Another observer included in the proposed virtual sensor is a band-pass torque observer that is used to distinguish between intentional human contact and accidental collision. In our previous work [14], we designed a second-order momentum deviation observer using the generalized momentum, and improved the performance of robot collision detection by adjusting the factor. This paper defines a new type of robot torque observation vector as
(15)$$r^{*}=K_{1}(\hat{P}-P)-\int (K_{2}r^{*}+K_{3}\int r^{*}dt)dt.$$

Similarly, from Equation (9), we have
(16)$$\dot{\hat{P}}=\tau -h(q,\dot{q})-r^{*}.$$

Substituting Equation (16) into Equation (15) yields
(17)$$r^{*}=K_{1}(\int (\tau -h(q,\dot{q})-r^{*})dt-P)-\int (K_{2}r^{*}+K_{3}\int r^{*})dt,$$
where $K_{1}$, $K_{2}$, and $K_{3}$ are diagonal positive definite matrices. The working principle diagram of the new torque observer is shown in Figure 3. 

The differential calculation of Equation (15) is
(18)$${\ddot{r}}^{*}=K_{1}{\dot{\tau }}_{f}-(K_{1}+K_{2})\dot{r}-K_{3}r.$$

The transfer function of the new type of torque observer can be obtained by Laplace transform as follows
(19)$$\frac{r_{i}^{*}(s)}{\tau_{ci}(s)}=\frac{K_{1i}s}{s^{2}+(K_{1i}+K_{2i})s+K_{3i}}.$$

Equation (19) shows that the new observer is a second-order system. Further analysis shows that Equation (19) is a standard band-pass filter transfer function, which can be rewritten as follows
(20)$$\frac{r_{i}^{*}(s)}{\tau_{ci}(s)}=\frac{A_{i}\xi_{i}\omega_{i}s}{s^{2}+\xi_{i}\omega_{i}s+{\left(\omega_{i}\right)}^{2}},$$
where $\omega_{i}$ is the center frequency of the band-pass filter, $\xi_{i}$ is the damping coefficient of the band-pass filter, $A_{i}$ is the gain of the band-pass filter, and $\xi_{i}\omega_{i}$ is the frequency bandwidth of the band-pass filter. Equation (19) can be determined by setting the relevant parameters in the transfer function (Equation (20)), so the new observer can be referred to as a band-pass torque observer (BPTOB).

In addition, when $K_{3}=K_{1}K_{2}$, Equation (19) can be further derived as
(21)$$\frac{r^{*}(s)}{\tau_{c}(s)}=\frac{K_{1}s}{\left(s+K_{1})(s+K_{2}\right)}=\frac{K_{1}}{s+K_{1}}\cdot \frac{s}{s+K_{2}}.$$

Thus, Equation (21) can be further represented by the multiplication of a first-order low-pass filtering (LPF) and a first-order high-pass filtering (HPF). By appropriately selecting the parameters of the LPF and the HPF, the detection of an accidental collision between the human and the robot can be achieved. In this paper, Equation (21) is used to simplify the calculation. Similarly, using the BPTOB to detect the external collision, we obtain
(22)$$r^{*}=[r_{1}^{*}\cdots r_{i}^{*},r_{i+1}^{*}\cdots r_{n}^{*}].$$

Comparing from $r_{n}^{*}$ to $r_{1}^{*}$ one by one with the joint threshold $N_{i}^{*}$, when the i link is involved in a collision, we obtain
(23)$$\left|r_{i}^{*}\right|>N_{i}^{*}\mathrm{and}\;\left|r_{i+1}^{*}\right|<N_{{}_{i+1}}^{*},\;i=1,2,\cdots n.$$

Simultaneously, the direction of the collision link can be determined by $\frac{r_{{}_{i}}^{*}}{\left|r_{{}_{i}}^{*}\right|}$. Otherwise, it is determined that no collision occurs. Therefore, the BPTOB can also accurately detect the collision information of the robot.

### 2.5. Implementation of the Virtual Sensor in a Robot System

As mentioned in the previous section, the two torque observers in the proposed virtual sensor—LPTOB and BPTOB—need to acquire the position of each joint of the robot and the output current information of the joint motor. However, their theoretical calculation formula is related to time. In practical applications, the robot control system samples discrete data at a certain frequency. Therefore, it is necessary to further derive the discretization calculation equation suitable for the robot system. 

Firstly, the differential calculation of the theoretical Equation (15) of the BPTOB can be expressed as
(24)$${\dot{r}}^{*}=K_{1}(\tau_{c}-r^{*})-(K_{2}r^{*}+K_{3}\int r^{*}dt).$$

The discretization calculation of the time integral of $r^{*}$ can be obtained as follows
(25)$$\int r^{*}dt=T\sum_{m=0}^{k}r^{*}(m),$$
where k is the discrete index, T is the sampling time of the robot control system, and m is an integer. The discrete equation for the forward difference of ${\dot{r}}^{*}$ is
(26)$${\dot{r}}^{*}=\frac{r^{*}(k+1)-r^{*}(k)}{T}.$$

Substituting Equations (25) and (26) into Equation (24) yields
(27)$$\frac{r^{*}(k+1)-r^{*}(k)}{T}=K_{1}[\tau_{c}(k)-r^{*}(k)]-K_{2}r^{*}(k)-K_{3}\sum_{m=0}^{k}r^{*}(m)].$$

Summing over $r^{*}(k)$ on both sides of the above equation gives
(28)$$\sum_{k=0}^{n}\left[r^{*}(k+1)-r^{*}(k)\right]=K_{1}T\sum_{k=0}^{n}\left[\tau_{c}(k)-r^{*}(k)\right]-K_{2}T\sum_{k=0}^{n}r^{*}(k)-K_{3}T\sum_{k=0}^{n}\sum_{m=0}^{k}\left[r^{*}(m)\right],$$
where n is an integer, setting the initial value of the observer as $r^{*}(0)=0$. Then, Equation (28) becomes
(29)$$r^{*}(n+1)=K_{1}T\sum_{k=0}^{n}\left[\tau_{c}(k)-r^{*}(k)\right]-K_{2}T\sum_{k=0}^{n}r^{*}(k)-K_{3}T\sum_{k=0}^{n}\sum_{m=0}^{k}\left[r^{*}(m)\right].$$

Substituting Equation (7) into the above equation yields
(30)$$r^{*}(n+1)=K_{1}T\sum_{k=0}^{n}\left[\tau (k)-h(q,\dot{q})(k)-\dot{P}(k)\right]-(K_{1}+K_{2})T\sum_{k=0}^{n}r^{*}(k)-K_{3}T\sum_{k=0}^{n}\sum_{m=0}^{k}\left[r^{*}(m)\right].$$

The discrete equation of the backward difference of $\dot{P}(k)$ is
(31)$$\dot{P}(k)=\frac{P(k)-P(k-1)}{T}.$$

Setting the initial value of the robot general momentum as P(0)=0, and after Equation (31) is substituted into Equation (30) and simplified, the final discrete calculation equation of the BPTOB can be obtained as follows
(32)$$r^{*}(n+1)=K_{1}T\sum_{k=0}^{n}\left[n_{har}T_{m}I_{m}(k)-h(q,\dot{q})(k)\right]-K_{1}P(n)-(K_{1}+K_{2})T\sum_{k=0}^{n}r^{*}(k)-K_{3}T^{2}\sum_{k=0}^{n}\sum_{m=0}^{k}\left[r^{*}(m)\right].$$

According to the above calculation and derivation process, the discrete calculation equation of the LPTOB can also be obtained as follows
(33)$$r(n+1)=KT\sum_{k=0}^{n}\left[n_{har}T_{m}I_{m}(k)-h(q,\dot{q})(k)\right]-KT\sum_{k=0}^{n}r(k)-KP(n).$$

### 2.6. Collision Detection and Distinction Process in the Virtual Sensor

During human–robot cooperation, the robot needs to observe the external force in real-time and judge the current state to determine the working mode. Based on the above analysis and the actual application requirements of the robot, the robot operating state is divided into three cases: (1)highest priority—collision(2)medium priority—contact(3)low priority—normal work

This robot state grading method can ensure the safety of humans and robots during the work, and can also fully take into account the interaction between the human and the robot as well as the independent operation of the robot. In addition, since there may be other disturbances during the operation of the robot, the method of distinguishing collision, contact, and disturbance by virtual sensors is as follows
$IF:\left|r_{i}^{*}\right|\geq N_{i}^{*}\to$ Collision$Else\;IF:\left|r_{i}^{*}\right|<N_{i}^{*}and\left|r_{i}\right|\geq N_{i}$$Then:IF:t\geq T_{thre}\to$ Contact, $Else:t<T_{thre}\to$ Disturbance$Else:\left|r_{i}^{*}\right|<N_{i}^{*}and\left|r_{i}\right|<N_{i}\to$ Normal work
where $T_{thre}$ is the time threshold. The thresholds $N_{i}^{*}$, $N_{i}$, and $T_{thre}$ can be obtained experimentally. The process of using virtual sensors to detect and distinguish contact and collision between humans and robots is shown in Figure 4.

Additionally, when the control torque of the joint motor driver changes in real-time during the process of the robot changing from a stationary position to movement, the position and velocity of the joint may be delayed due to the inertia of the robot. The following equation is used to correct r and $r^{*}$.
(34)Δr=α⋅Φ+β⋅Θ,
where Δr is the corrected value, α and β are the adjustment factors, Φ is the parameter related to the joint motor and servo drive, and Θ is the inertia-related parameter of the robot. When the robot changes from stationary to motion modes, α and β are positive in the correction equation for the observed torque, while the other motion states are zero. 

## 3. Friction Identification and Compensation

In order to reduce the influence of friction on the virtual sensor, this paper improves the sensitivity of the torque observer by identifying and compensating the joint friction [20,21]. Kennedy et al. pointed out that the Stribeck friction model is suitable for describing the conventional industrial robot joint friction [22], and its basic expression can be expressed as
(35)$$F(\dot{q})=F_{c}\mathrm{sgn}(\dot{q})+F_{s}e^{-(\dot{q}/{\dot{q}}_{s})\sigma }\mathrm{sgn}(\dot{q})+F_{v}\dot{q},$$
where $F_{s}$ is the Stribeck parameter, ${\dot{q}}_{s}$ is the Stribeck velocity, $F_{c}$ is the Coulomb friction coefficient, $F_{c}\mathrm{sgn}(\dot{q})$ is the Coulomb friction, $F_{v}$ is the viscous friction coefficient, $F_{v}\dot{q}$ is the viscous friction, and σ is the constant associated with the contact surface geometry.

In order to facilitate data processing with MATLAB software, Equation (35) is simplified to
(36)$$F(\dot{q})=\lambda_{1}+\lambda_{2}\exp(-\lambda_{3}\dot{q})+\lambda_{4}\dot{q},$$
where $\lambda_{1}$, $\lambda_{2}$, $\lambda_{3}$, and $\lambda_{4}$ are the required parameters. Then, the parameters in Equation (36) are estimated offline using the least squares method. 

In this paper, independent experiments were performed on each joint to obtain the friction force of the joint and the parameters of the Stribeck friction model. During the experiment, one of the joints of the robot was controlled to move at a constant speed $\ddot{q}=0$, and the other joints were fixed. When no external force is applied to the robot, from Equations (1) and (4), we have
(37)$$g(q)+\tau_{f}=n_{har}T_{m}I_{m}.$$

In Equation (37), the joint control torque only contains gravity and friction torque. The joint of the robot is controlled to move in a positive and negative direction within a given interval. At this point, Equation (37) becomes
(38)$$\sum (g{\left(q\right)}^{+}+\tau_{f}^{+})=\sum {\left(n_{har}T_{m}I_{m}\right)}^{+},$$
(39)$$\sum \left(g{\left(q\right)}^{-}+\tau_{f}^{-}\right)=\sum {\left(n_{har}T_{m}I_{m}\right)}^{-},$$
where superscript symbols “+” and “−” indicate positive and negative directions, respectively. When the joint is in the same position during positive and negative motion, the gravity has $g{\left(q\right)}^{+}=g{\left(q\right)}^{-}$ and the friction has $\tau_{f}^{+}=-\tau_{f}^{-}$. Subtracting Equation (38) from Equation (39) gives
(40)$${\bar{\tau }}_{f}=\frac{\sum {\left(n_{har}T_{m}I_{m}\right)}^{+}-\sum {\left(n_{har}T_{m}I_{m}\right)}^{-}}{2A},$$
where A is the amount of data collected in a single-speed experiment. Therefore, the optimized friction parameters can be obtained by the following formula.
(41)$$\min\Lambda {\sum_{1}^{\Gamma }\left[{\bar{\tau }}_{f}-(\lambda_{1}+\lambda_{2}\exp(-\lambda_{3}\dot{q})+\lambda_{4}\dot{q})\right]}^{2},$$
where Γ represents the friction data at different speeds. The above experimental method can reduce the influence of gravity and torque measurement errors on the joint friction measurement, and obtain more accurate friction model parameters. 

## 4. Experimental Results

### 4.1. Comparison of Different Force Signal Distribution Characteristics

In this paper, experimental methods were used to compare and analyze the distribution characteristics of different force signals. In the experiment, the robot end with an ATI force sensor was directly pushed by a human hand, and the intentional human contact force data was acquired in real-time through the upper computer at a sampling frequency of 125 Hz. In order to make the data universal, five lab members each repeatedly pushed the robot five times. In addition, a rubber hammer was used to strike the end of the robot to simulate an accidental collision, and the ATI force sensor as also used to collect the collision force data in real-time. The intentional human contact force and collision force were distributed in the time domain as shown in Figure 5. The contact force and the collision force shown in Figure 5 were analyzed offline by Fourier-transform using MATLAB software, and the corresponding spectral distribution was obtained as shown in Figure 6. 

As shown in Figure 5, the rising process of the intentional human contact force is relatively stable and lasts for a long time, while the collision force has a large mutation and a short duration. Figure 6 shows that the high amplitude of the intentional contact force signal is concentrated in the low-frequency region, while the collision force signal amplitude distribution is wider. The shorter the duration of the collision force, the wider the range of the amplitude frequency distribution. In addition, the magnitude of the critical threshold $w_{1}$ in Figure 1 determines the observation effect of the BPTOB. If the threshold is large, the robot will be insensitive to a slight collision. According to the results of Refs. [16,17,18], the value of 4.6 Hz can be used as the threshold for the BPTOB to filter low-frequency signals. The upper threshold in the virtual sensor was taken as 19.5 Hz to filter out interference from high-frequency signals in the system. 

### 4.2. Friction Identification and Compensation Experiment

Experimental verification was carried out using a six-DOF conventional industrial robot. The overall structure of the robot system is shown in Figure 7. The control cabinet contained a GALIL DMC motion control card and a joint motor servo driver. The system control software was run on the computer for robot motion planning and virtual sensor implementation. It used Ethernet to communicate with the DMC motion control card in the control cabinet. The motion control card controlled the servo driver, which controlled the motion of the robot. The motor servo driver and the computer communicated via the serial port. The control software on the computer could directly set the state of the servo driver through the serial port and read the relevant parameters of the servo driver.

In the first experiment, taking joint 3 as an example, the joint friction identification and compensation was performed using the method described in Section 3. The robot joint 3 was controlled to move at different speeds in the range $\left[-60^{{}^{\circ}},30^{{}^{\circ}}\right]$. After filtering out the singular points of the collected data, 40 sets of effective data were obtained in the speed range $0-20^{{}^{\circ}}\cdot s^{-1}$ and the friction fitting results are shown in Figure 8. The fitting parameters are listed in Table 1. Similarly, the friction parameters of other joints can be obtained experimentally.

In order to simplify the calculation, the second and third links of the robot were selected for experimentation. The specific implementation process of the virtual sensor in the robot system is detailed in [14]. The Stribeck model was used to compensate the joint friction to improve the detection performance of the LPTOB. This controlled the second and third joint movements of the robot while simultaneously running the virtual sensor. Controlling the movement of the second and third joints of the robot and running the virtual sensor at the same time, the experimental results obtained when no external force was applied are shown in Figure 9.

Figure 9a shows the motion parameters of the robot when there is no external force. In Figure 9b, it can be seen that the model errors and disturbances cause deviations in the observer’s observations. It is thus necessary to set a reasonable threshold to avoid false detection. When there is no friction compensation, the BPTOB observation error is small and the LPTOB observation error is very large. The experimental results show that the joint friction affects the LPTOB observation, but the effect on BPTOB observation is not obvious. Therefore, it is necessary to reduce the threshold of collision detection of the LPTOB by friction compensation, as shown in Figure 9c. Moreover, the experiments showed that the interference signals generated by the robot system during operation are mainly low-frequency signals, which provides the basis for the realization of the proposed virtual sensor. Based on the above experiments and analysis, after the observation results were corrected by Equation (34), the thresholds of joints 2 and 3 in the LPTOB were set to 9 N.m and 6 N.m, respectively. 

### 4.3. Observer Detection Performance Experiment

In the second experiment, the robot was controlled to move from the starting position $\left[40^{{}^{\circ}},-50^{{}^{\circ}}\right]$ to the ending position $\left[90^{{}^{\circ}},10^{{}^{\circ}}\right]$. During the movement of the robot, a lab member intentionally applied a force to link 2 from the forward and reverse directions, and then used a rubber hammer to strike link 2 from the forward and reverse directions to simulate a collision. The output torque of the joint and the observation result of the LPTOB are shown in Figure 10. The BPTOB observation result is shown in Figure 11. Then, the intentional contact force and the collision force were similarly applied to link 3, and the experimental results are shown in Figure 12 and Figure 13, respectively.

As can be seen from Figure 10, the output torque of joint 2 changes correspondingly when link 2 is subjected to external forces. The LPTOB can detect the magnitude and direction of the external force of joint 2 in real-time. Joint 3 is not subjected to external force, and the observation value of the LPTOB is small. The observation results are consistent with Equation (2). Figure 11 shows that the BPTOB is more sensitive to the collision force signal, and can attenuate components of the force signal with frequencies below 4.6 Hz. Comparing the experimental results shown in Figure 10 and Figure 11, by setting a reasonable threshold of the BPTOB and using the judgment method described in Section 2.6, the robot can distinguish between intentional contact and accidental collision. Similarly, as shown in Figure 12 and Figure 13, when link 3 is subjected to external forces, all external forces can be detected in real-time by the LPTOB. Meanwhile, the BPTOB detects only collision forces having higher frequency signals. At the same time, the BPTOB observations also change greatly when the contact force is large, because the intentional human contact force also contains high-frequency components. In this paper, the BPTOB threshold was set to 6.5 N·m throughout the experiments. In practical applications, the BPTOB threshold should be set as low as possible to avoid skin abrasion and contusion caused by slight collision forces. The above experimental results show that the proposed virtual sensor uses the LPTOB and BPTOB to detect external forces simultaneously, which can effectively distinguish between intentional human contact and accidental collision.

Another experiment was conducted to compare the proposed method that reported in Ref. [16]. In order to compare the performance and differences between the two methods, the cut-off frequency parameters obtained in Section 4.1 were used in the experiments. To avoid the duplication of content, the theory and implementation of the two methods can be found in Section 2 of this paper and Section 3 of Ref. [16]. The dynamic model includes friction compensation to reduce the influence of model errors. Since the observation result of the observer in virtual sensors is related to the joint torque, equivalent torque filtering is also used when implementing the method reported in Ref. [16]. At the same time, the output torque of the joint is filtered by a low-pass filter to avoid the influence of other unknown interferences on the comparison experiment. These two methods are called the momentum-based method and the signal-based method in this paper. Without loss of generality, this paper considers the scenario in which the robot is in normal motion mode, different external forces are applied to link 2 and both methods are run simultaneously. The experimental results of the two methods are shown in Figure 14.

Figure 14a shows the real-time motion parameters of the robot in the comparative experiment. From the experimental results shown in Figure 14b, we can see that the LPTOB observations and low-pass filtering results are generally similar. Both can effectively detect external forces. As shown in Figure 14c, the high-pass filtering result is susceptible to residual torque (or current), resulting in a certain offset. The BPTOB observations fluctuate mainly around zero, which shows that it has better anti-interference ability. At the same time, the BPTOB is only sensitive to high-frequency signals. Moreover, when the external force changes greatly, the high-pass filtering result is prone to delay. This is because the signal-based filtering method requires the use of several previous residual torque (or current) measurements. The momentum-based approach provides the better real-time observation of external forces. At the same time, the momentum-based method does not require acceleration information or inverse dynamics calculations, which can reduce errors. 

### 4.4. Collision Detection and Distinction Experimentusing the Virtual Sensor

In this section, the proposed virtual sensor was applied to a workplace where humans and robots cooperate to further verify its performance in an actual working environment. A screenshot of the experimental results is shown in Figure 15. The robot motion parameters during the experiment are shown in Figure 16. The observations of the BPTOB and LPTOB are shown in Figure 17 and Figure 18, respectively. 

The initial position of the robot is shown in Figure 15a. Figure 15b shows the robot operating normally at a given planned trajectory and speed. At this time, the observations of the BPTOB and LPTOB in Figure 17 and Figure 18 are both smaller than the threshold. Figure 15c shows the human and the robot sharing workspace. In order to complete some delicate and complicated work in human–robot cooperation, such as adjusting the workpiece or adding materials, the operator needs to drag the robot directly. As shown in Figure 15d, the human applies an intentional contact force on the robot, and the force is controlled by the human without causing danger. At this time, the observed values of the BPTOB in Figure 17 are smaller than the threshold. The observation value of the LPTOB in Figure 18 changes in real-time with the contact force, and the observation result coincides with the change of the output torque of the robot joint in Figure 16. Therefore, the robot can detect the intentional contact force and suspend the current work. When the robot detects that the external contact force has disappeared, it waits 2 seconds to ensure that the human reaches a safe area and continues to work, as shown in Figure 15e,f. Once the robot detects an accidental collision with the dummy in the workspace, as shown in Figure 15g, the BPTOB observation immediately mutates and rapidly rises above the threshold value, as shown in Figure 17, which is consistent with the variation of joint output torque in Figure 16. The robot immediately takes safety measures to move the colliding link in the opposite direction of the collision to ensure the safety of the human, as shown in Figure 15h,i. 

## 5. Conclusions

The robot can independently detect and distinguish intentional human contact from accidental collision between a human and the robot, which is of great significance for improving the adaptability of the robot in complex environments, ensuring human safety and improving the information interaction efficiency. This paper presents a virtual sensor for collision detection and differentiation. Two observers are designed in this virtual sensor, named the LPTOB and the BPTOB. Based on the different frequency domain distribution characteristics of the force signals in contact and collision, the detection and distinction of contact and collision between humans and robots are realized by using the unique filtering system in the LPTOB and the BPTOB. The virtual sensor does not require acceleration information or inverse dynamics calculations, which can reduce errors and calculations. It only needs the drive current and position information of the robot joint motor, which is suitable for conventional industrial robots and reduces costs. Force detection experimental results show that the proposed virtual sensor can distinguish the physical contact types on the robot end point and the body, and has stable work performance and high reliability. Compared with the existing methods, the proposed method has better observation performance for external forces. This is of great significance for the real-time detection of accidental collisions in human–robot interaction and for the safety of humans and robots. In addition, the virtual sensor was further validated by human and robot interaction experiments. Subsequent research will improve the detection accuracy of the external force of the virtual sensor through dynamic parameter identification. It should be pointed out that actual contact and collision between humans and robots often have a certain randomness. The proposed virtual sensor is rarely 100% accurate, so it can only be used as an important judgment and reference tool for distinguishing between contact and collision. Particularly for human contact, the distinction between rapid intentional contact and slow unintentional contact should also be addressed in further research.

## Figures and Tables

**Figure 1 sensors-19-02368-f001:**
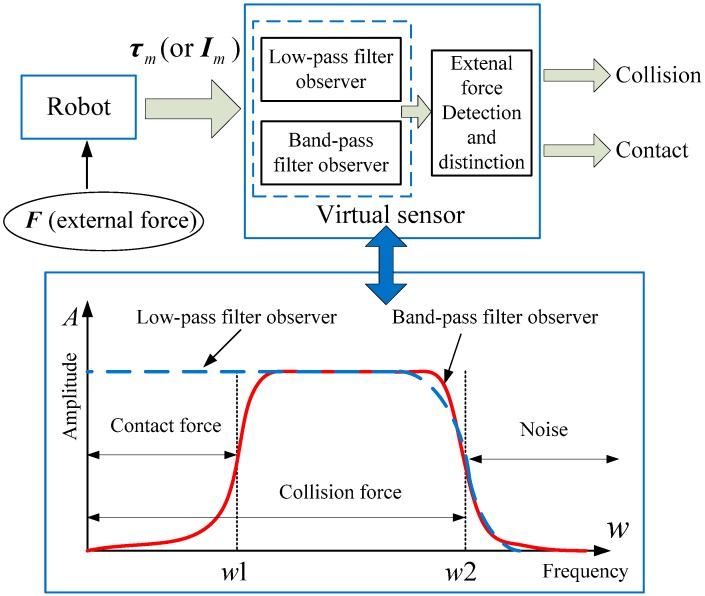
The proposed virtual sensor scheme.

**Figure 2 sensors-19-02368-f002:**
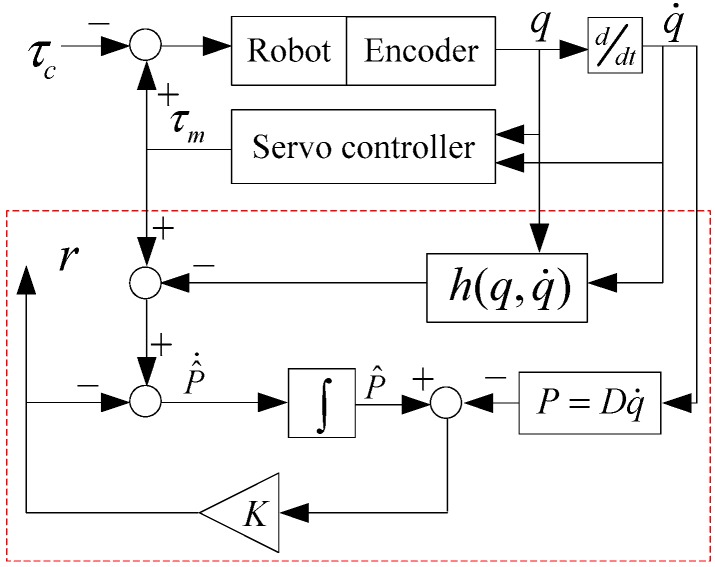
Working principle diagram of the momentum deviation observer.

**Figure 3 sensors-19-02368-f003:**
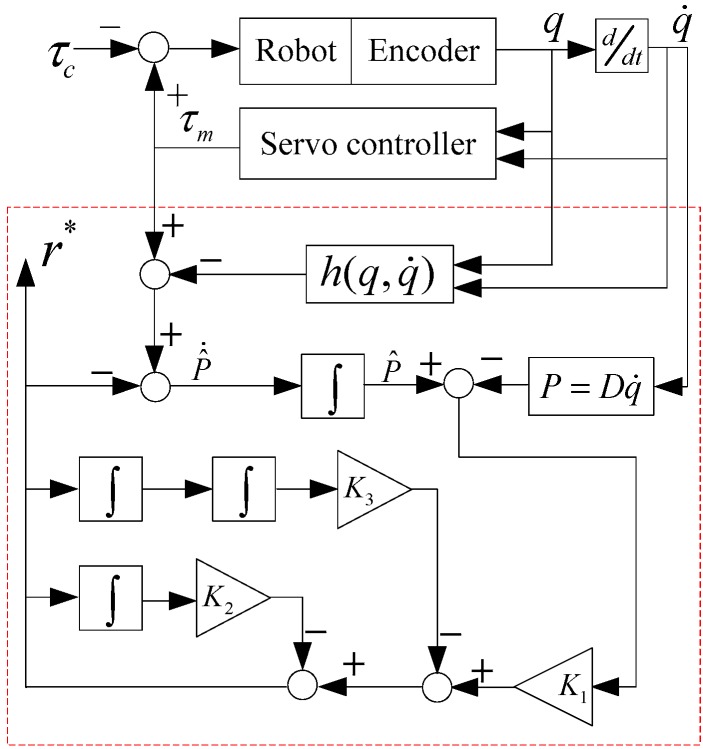
Working principle diagram of the new torque observer.

**Figure 4 sensors-19-02368-f004:**
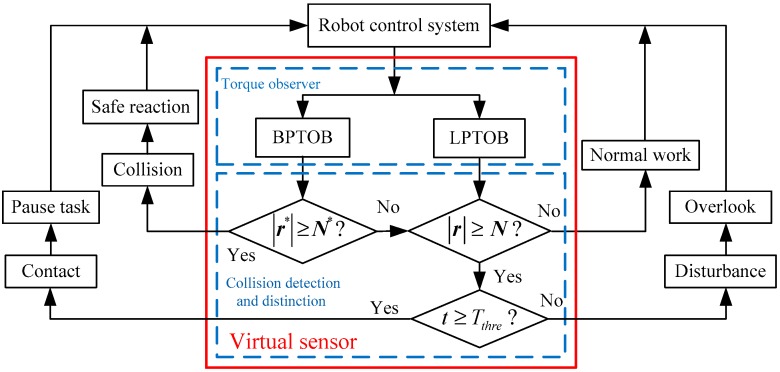
Flowchart of collision detection and distinction by a virtual sensor.

**Figure 5 sensors-19-02368-f005:**
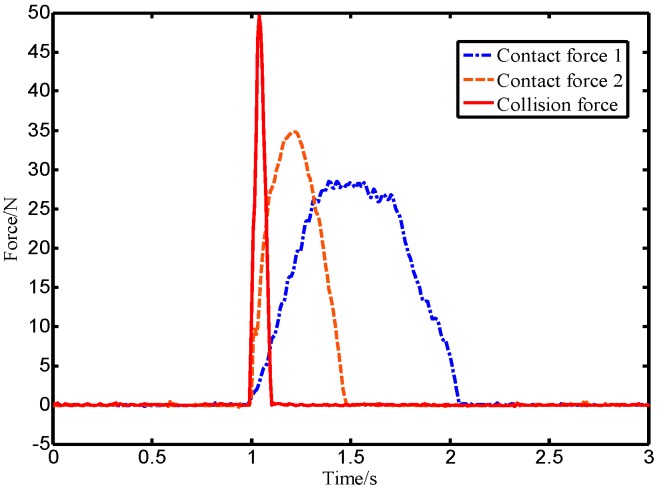
Time domain distribution diagram of force signals.

**Figure 6 sensors-19-02368-f006:**
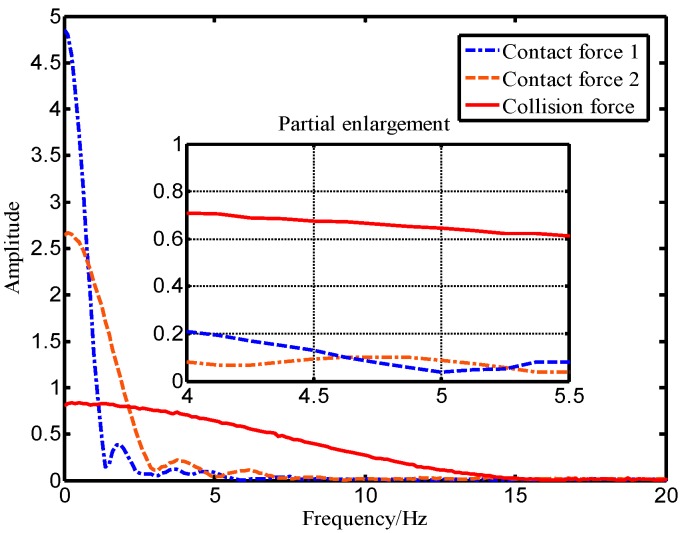
Frequency domain distribution diagram of force signals.

**Figure 7 sensors-19-02368-f007:**
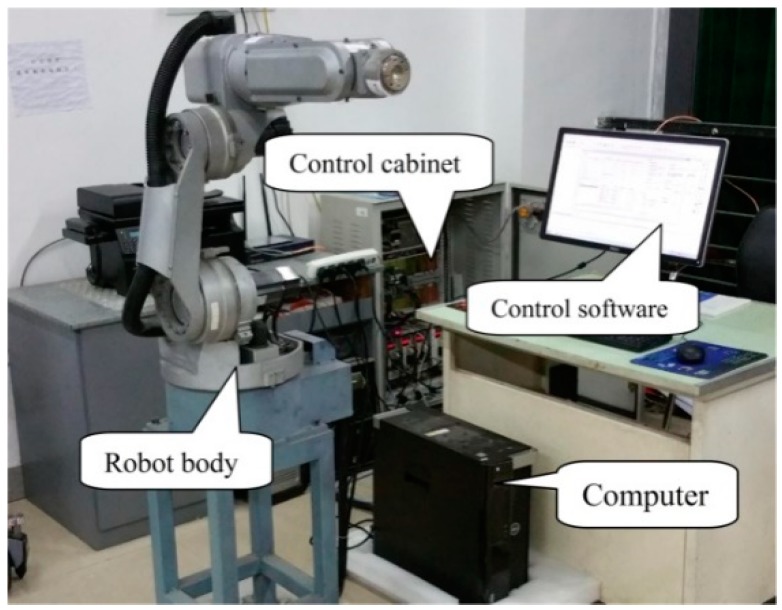
Robot experimental platform.

**Figure 8 sensors-19-02368-f008:**
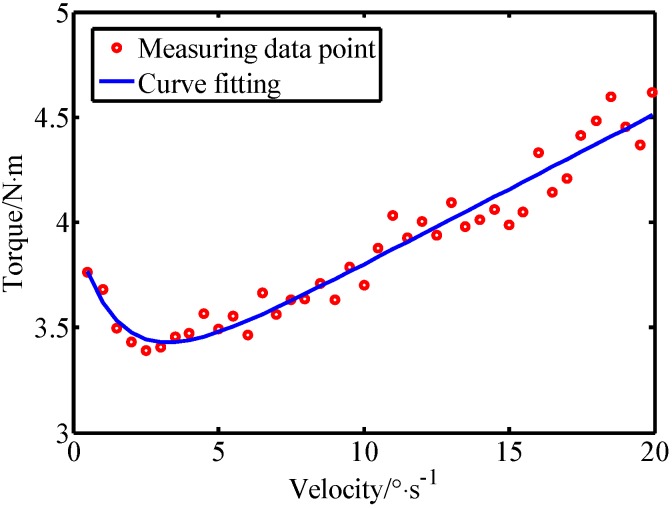
The fitting result of the Stribeck model.

**Figure 9 sensors-19-02368-f009:**
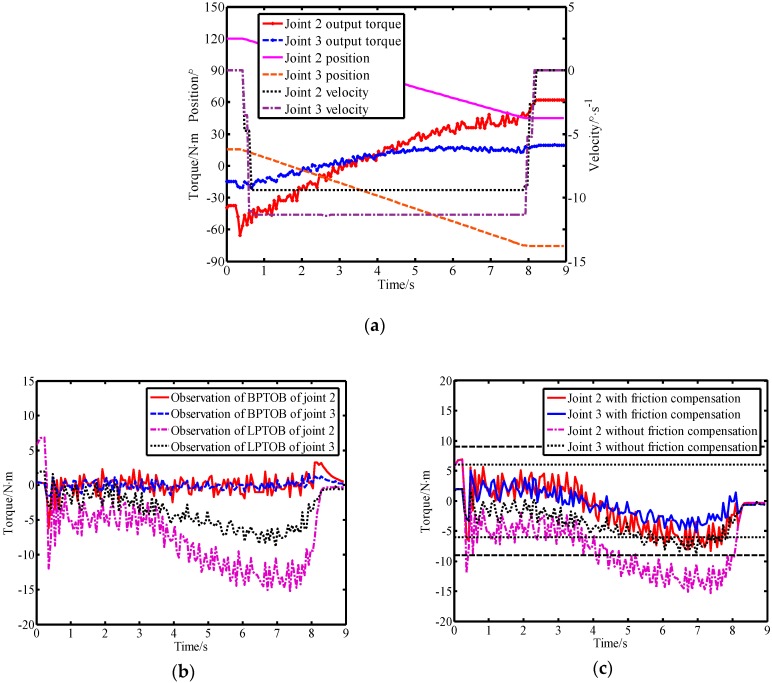
Experimental observations and comparisons. (**a**) Robot motion parameters, (**b**) comparison of observation results of the band-pass torque observer (BPTOB) and the low-pass torque observer (LPTOB), and (**c**) LPTOB observations with or without friction compensation.

**Figure 10 sensors-19-02368-f010:**
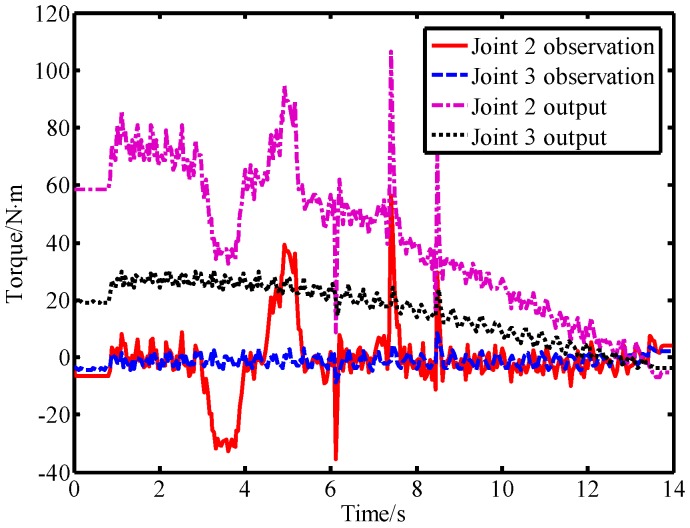
Observations of the LPTOB and joint output torque when force is applied on link 2.

**Figure 11 sensors-19-02368-f011:**
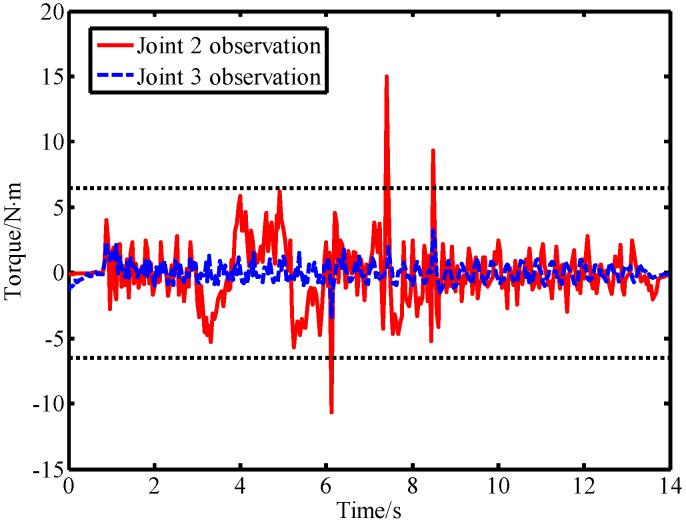
Observations of the BPTOB when force is applied on link 2.

**Figure 12 sensors-19-02368-f012:**
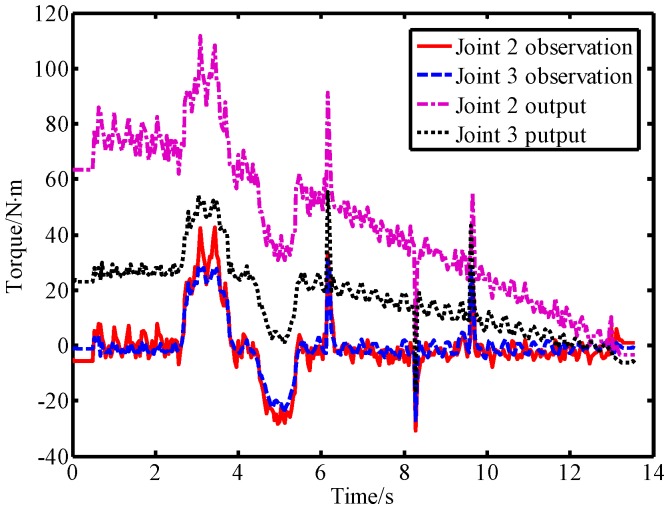
Observations of the LPTOB and joint output torque when force is applied on link 3.

**Figure 13 sensors-19-02368-f013:**
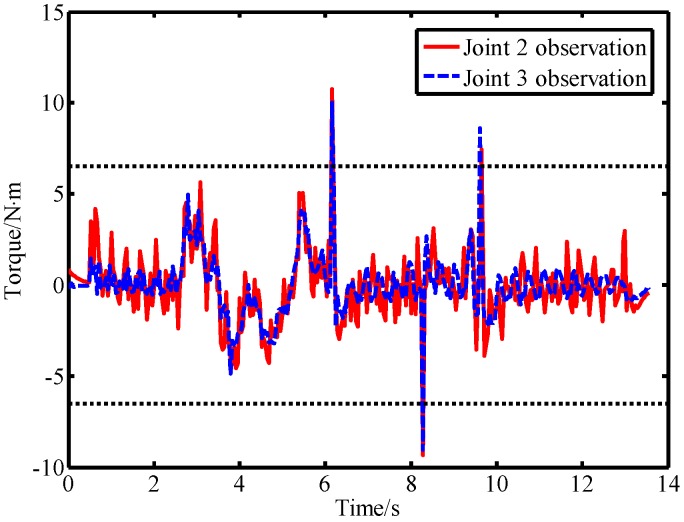
Observations of the LPTOB when force is applied on link 3.

**Figure 14 sensors-19-02368-f014:**
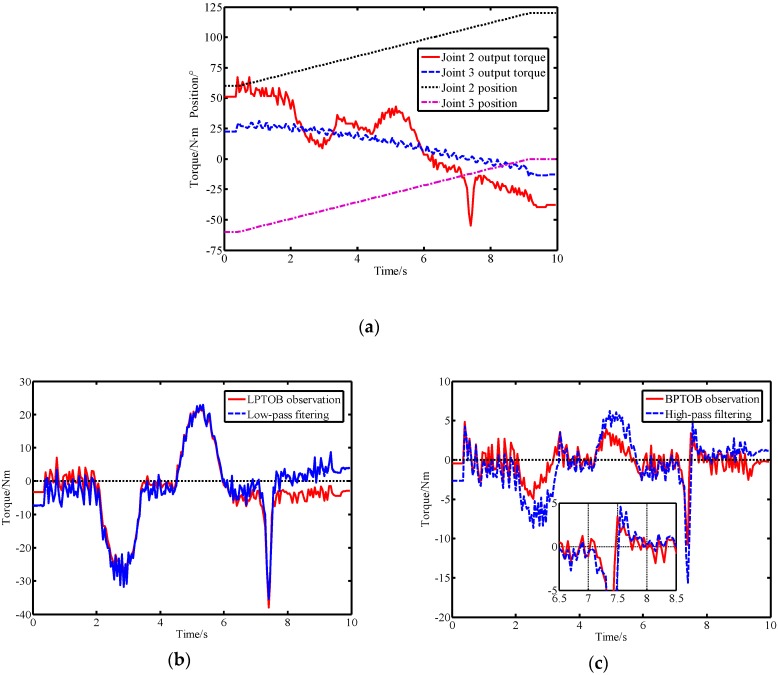
Experimental comparison of the momentum-based method and the signal-based method. (**a**) Motion parameters of the robot when subjected to external forces, (**b**) observations of the LPOB and low-pass filtering results of the residual torque, and (**c**) observations of the BPOB and high-pass filtering results of the residual torque.

**Figure 15 sensors-19-02368-f015:**
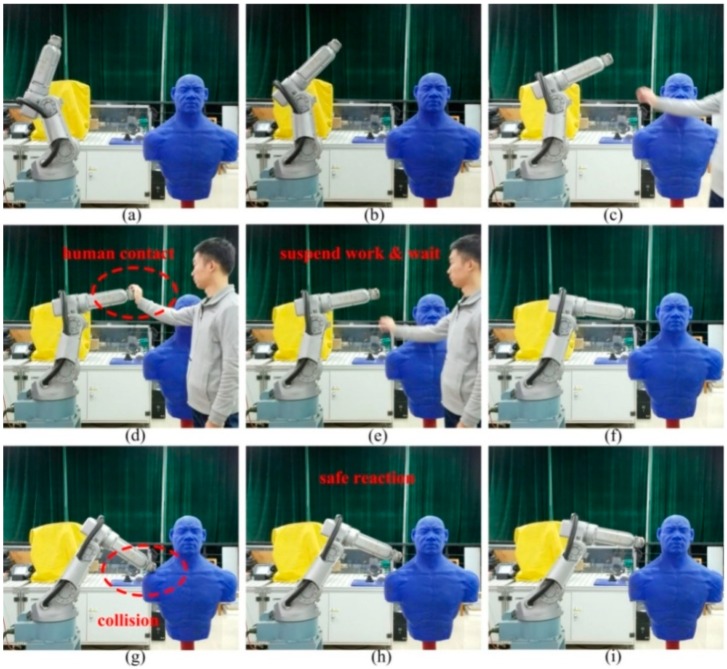
Human–robot interaction experiment for distinction between contact and collision. (**a**) The initial position of the robot, (**b**) the robot is operating normally, (**c**) the human and the robot sharing workspace, (**d**) the human applies an intentional contact force on the robot, (**e**) the external contact force has disappeared, (**f**) the robot continues to work, (**g**) the robot detects an accidental collision, (**h**) the robot takes safety measures to ensure the safety of human, (**i**) the robot stop position.

**Figure 16 sensors-19-02368-f016:**
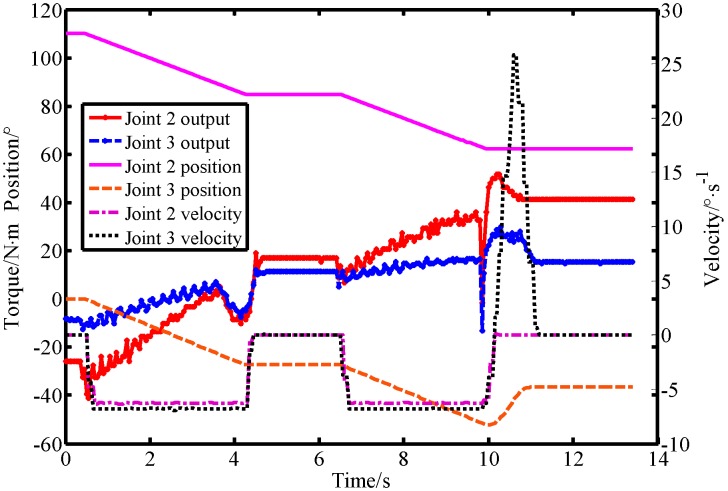
Robot motion parameters in human–robot interaction experiments.

**Figure 17 sensors-19-02368-f017:**
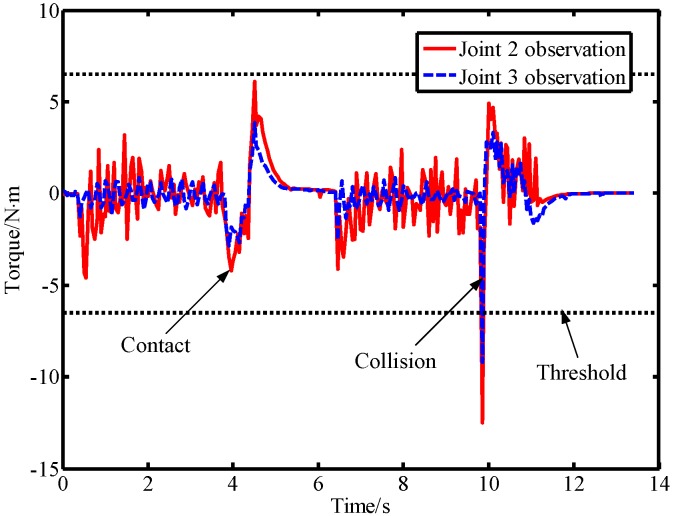
Observations of the BPTOB in human–robot interaction experiments.

**Figure 18 sensors-19-02368-f018:**
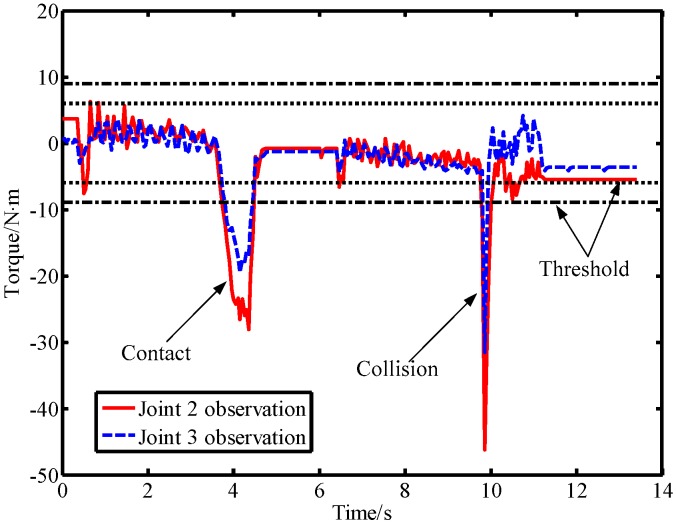
Observations of the LPTOB in human–robot interaction experiments.

**Table 1 sensors-19-02368-t001:** Stribeck curve fitting coefficient.

Parameter	$\lambda_{1}$	$\lambda_{2}$	$\lambda_{3}$	$\lambda_{4}$	Resnorm
Joint 3	3.0848	0.8797	0.6312	0.0715	0.2783

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
