# Peer review of "A Virtual Sensor for Collision Detection and Distinction with Conventional Industrial Robots"

_sensors, 2019, doi:10.3390/s19102368_

Reviewer 1 Report

The paper is interestimg and well structured. Perhaps it could be shortened a little bit and focus on relevant contributions.

A comparison with other systems should be provided in order to show the differences and improvements..

Results section need some more explanations and figures could be difficult to follow. 

Conclusions should focus in novelties and usefulness of the proposed solution.

Author Response

Dear Reviewers:

Thank you very much for your careful review and constructive suggestions with regard to our manuscript “A virtual sensor for collision detection and distinction with conventional industrial robots”. Those comments are so valuable and helpful for authors to correct and improve the manuscript. We have studied comments carefully and tried our best to revise the manuscript and made corresponding changes in the manuscript. Appended to this letter is our point-by-point response to the comments that you raised.

Reviewer:

Reviewer’s Comments

1. The paper is interesting and well structured. Perhaps it could be shortened a little bit and focus on relevant contributions.

Response 1: Thanks for your valuable suggestion. We have simplified some of the content, improved the quality of the manuscript and highlighted the contributions of our manuscript.

2. A comparison with other systems should be provided in order to show the differences and improvements.

Response 2: We have add an experiment to compare the method proposed in our manuscript with the method proposed in references [16-17], highlighting the differences and advantages of the method proposed in our manuscript. Please see pages 18-19 of Section 4.3 of the manuscript.

3. Results section need some more explanations and figures could be difficult to follow.

Response 3: We apologize for the trouble that makes it difficult for you to read and understand. We have further improved the analysis of the results of the experimental part. We hope to meet your requirements.

4. Conclusions should focus in novelties and usefulness of the proposed solution.

Response 4: Thanks for your valuable suggestion. We have revised the conclusions accordingly, highlighting the novelty and usefulness of the proposed method.

Reviewer 2 Report

The paper proposes a virtual sensor for collision detection and distinction, whose key points are:

-          distinction between robot physical interactions with humans and unexpected collisions through the analysis of the frequency distribution ranges of the force (and then of the torques and currents on the joints) in the two cases;

-          adoption of two observers acting as proper filters to distinguish the two situations;

-          adoption of constant thresholds on the external momentum to distinguish and switch between the different robot operating states (i.e., collision, contact, normal working);

-          friction identification to improve the torque computation provided by the robot dynamic model.

The basic concepts of the proposed approach are very similar (but with more limited results in the implementation, in my opinion) to those of the “Virtual Sensors for Robot Manual Guidance and Collision Detection” approach developed in Section 2 of the following (not cited) paper, recently published just in the same journal:

M. Indri, L. Lachello, I. Lazzero, F. Sibona, S. Trapani, “Smart Sensors Applications for a New Paradigm of a Production Line”, Sensors, vol. 19, n.3, 650, 2019

In such a paper, the same kind of distinction is made on the basis of filters acting on different frequency ranges, even if not configured as observers. The approach developed in the above paper seems to be computationally lighter, and experimentally demonstrated to perform successful manual guidance sessions (and not only to simply distinguish the different situations). Moreover, it is intrinsically more robust than the proposed one, since time-varying thresholds are adopted, and the slope of the force is monitored, too, inside a complete finite state machine, which manages the transitions between the various recognized situations (similarly somehow to what the flowchart in Figure 4 of the submitted paper does). The use of constant thresholds, as in the approach developed in the submitted paper, should be avoided, since it increases the risk of false collision detection, because small changes in the measured motor currents (e.g., due to temperature) could be sufficient to determine wrong detections.

In addition, the inclusion of a proper, well identified friction component in the robot dynamic model is mandatory for any algorithm (not only aimed at collision detection) that somehow relies on the comparison of the measured motor currents (or torques) on the joints and the ones computed by the considered model of the manipulator. Section 3 does not offer any new original contribution about friction identification; also the procedure to remove the gravity components detailed in the manuscript is well known.

Last but not least, English should be improved.

Summarizing, since the paper does not offer a solid, original contribution, improving the current state-of-the-art, it is not suitable for publication in the current form, in my opinion. The authors should carefully compare the proposed solution with other ones already available in literature (not only that of the paper cited above) to prove that their approach provides actual advantages.

Author Response

Dear Reviewers:

Thank you very much for your careful review and constructive suggestions with regard to our manuscript “A virtual sensor for collision detection and distinction with conventional industrial robots”. Those comments are so valuable and helpful for authors to correct and improve the manuscript. We have studied comments carefully and tried our best to revise the manuscript and made corresponding changes in the manuscript. Appended to this letter is our point-by-point response to the comments that you raised.

Reviewer:

the response letter by the authors only partially addresses my concerns, since they correctly highlight differences and some advantage of their solution, but not some drawbacks that I indicated in my review. In addition, nothing is said about the friction issue.

- the key points of the authors' response letter

- an accurate analysis of both pros and cons of the proposed approach (in particular with reference to computational burden and robustness

- a complete reply to my previous review)

Dear reviewer, thank you very much for your careful review. Although the momentum-based algorithm in our manuscript does not require secondary derivation to calculate acceleration and does not involve inverse dynamics calculations, However, due to the low-pass and band-pass observers, there are momentum calculations, and the overall calculation is indeed relatively large. In our manuscript, the constant threshold method is used for collision detection. In theory, it is not as good as the time-varying threshold method, and the robustness will be worse. The purpose of using friction compensation in our method is to improve the accuracy of the dynamic model, thereby improving the accuracy of collision detection and reducing the adverse effects of constant thresholds. The advantages of our method are as follows:

1. The method proposed in our manuscript does not need to calculate the acceleration of each joint of the robot. This is very important because the acceleration of the robot is usually obtained by the second differential of the joint displacement. The calculation error is very large. Avoiding the acceleration calculation can greatly improve the accuracy of the collision detection.

2. The method proposed in our manuscript does not need to perform inverse dynamics calculations to simplify the calculation process.

3. In our manuscript, the momentum-based observers can obtain filtered results by sampling once in the time domain. However, signal filtering based on polynomial method needs to collect data several times before filtering can be realized, and there is a time delay. This is very important for collision detection.

Reviewer’s Comments

The paper proposes a virtual sensor for collision detection and distinction, whose key points are:

- distinction between robot physical interactions with humans and unexpected collisions through the analysis of the frequency distribution ranges of the force (and then of the torques and currents on the joints) in the two cases;

- adoption of two observers acting as proper filters to distinguish the two situations;

- adoption of constant thresholds on the external momentum to distinguish and switch between the different robot operating states(i.e., collision, contact, normal working);

- friction identification to improve the torque computation provided by the robot dynamic model.

1. The basic concepts of the proposed approach are very similar (but with more limited results in the implementation, in my opinion) to those of the “Virtual Sensors for Robot Manual Guidance and Collision Detection” approach developed in Section 2 of the following (not cited) paper, recently published just in the same journal:

M. Indri, L. Lachello, I. Lazzero, F. Sibona, S. Trapani, “Smart Sensors Applications for a New Paradigm of a Production Line”, Sensors, vol. 19, n.3, 650, 2019.

Response 1: Thank you very much for your careful review. As professor Indri's paper has just been published, we did not see it during research and manuscript writing. We have added and cited the paper as reference [17] in the revised manuscript.

2. In such a paper, the same kind of distinction is made on the basis of filters acting on different frequency ranges, even if not configured as observers. The approach developed in the above paper seems to be computationally lighter, and experimentally demonstrated to perform successful manual guidance sessions (and not only to simply distinguish the different situations). Moreover, it is intrinsically more robust than the proposed one, since time varying thresholds are adopted, and the slope of the force is monitored, too, inside a complete finite state machine, which manages the transitions between the various recognized situations (similarly somehow to what the flowchart in Figure 4 of the submitted paper does). The use of constant thresholds, as in the approach developed in the submitted paper, should be avoided, since it increases the risk of false collision detection, because small changes in the measured motor currents (e.g., due to temperature) could be sufficient to determine wrong detections.

Response 2: Thank you very much for your careful review. We appreciate that Professor Indri's paper proposed a good method to manage both collision detection and manual guidance sessions, which manages the transitions between the various recognized situations. Such an approach is defined as sensor-less since no external sensors are required, and only need the proprioceptive sensors of robot. It is very easy to implement on industrial robots. In addition, in the proposed approach, a complete finite state machine is included, which manages all the phases of the developed procedure, including Monitoring, Manual Guidance, Collision Reaction and Waiting four states. On the one hand, in monitoring phase, this method can detect if a collision occurred and distinguishes if it was due to an accidental impact with the environment, during a non-collaborative application, or determined by an intended human-robot contact. On the other hand, in the post-impact phase, this method can impose an appropriate reaction strategy: an MG algorithm when an intended human-robot contact is detected or a CD reaction when an accidental collision occurs. In Indri's paper, the principle and method of filtering are intuitive and easy for readers to understand. This method is simple and effective, and experimentally demonstrated to perform successful manual guidance sessions. This approach for manual guidance, increasing the potentialities of a standard industrial manipulator. Moreover, In Indri's paper, a acceleration based time varying thresholds was adopted to avoid wrong detections and improve the robust performance of algorithm. In our submitted manuscript, we mainly focused on collision detection and distinction.

3. In addition, the inclusion of a proper, well identified friction component in the robot dynamic model is mandatory for any algorithm (not only aimed at collision detection) that somehow relies on the comparison of the measured motor currents (or torques) on the joints and the ones computed by the considered model of the manipulator. Section 3 does not offer any new original contribution about friction identification; also the procedure to remove the gravity components detailed in the manuscript is well known.

Response 3: We agree with the reviewer that the dynamics model can reduce the error by friction compensation. Instead of delving into the principle of friction, we used experimental methods to obtain more realistic joint friction. This method avoids the interference of dynamic model error on joint friction measurement, and a more realistic friction torque value can be obtained. The friction compensation experiment shows that the joint friction of conventional industrial robots has a great influence on the collision detection. Friction compensation can reduce the model error and reduce the threshold value of the proposed method in our manuscript to improve the performance of collision detection.

4. Last but not least, English should be improved.

Response 4: Thank you very much. We have tried our best to modify the language and improve the quality of the manuscript.

5 Summarizing, since the paper does not offer a solid, original contribution, improving the current state-of-the-art, it is not suitable for publication in the current form, in my opinion. The authors should carefully compare the proposed solution with other ones already available in literature (not only that of the paper cited above) to prove that their approach provides actual advantages.

Response 5: Thanks for your careful review and valuable suggestion. Our work may not meet the requirements of the reviewer. We think that method proposed in our manuscript is different from the method proposed in references [16-17] (Geravand's paper and Professor Indri's paper). In our manuscript, a momentum-based method was adopted to design a virtual sensor to realize collision detection and distinction. Our method does not need acceleration and inverse dynamic calculation, which reduces the source of error and simplifies the calculation process. At the same time, our method has better real-time observation performance for external forces.

Reviewer 3 Report

 The authors presents a virtual sensor for collision detection and distinction in the human-robot interaction process.

- The idea is interesting and somewhat practical in the industrial environment.

- The paper is well-organized and written. 

- The result is supported by the experiments and analysed well.

- The limitations and future work is properly outlined.

Nevertheless, the authors are encouraged to compare their existing work with others, and also demonstrate why virtual sensor implementation is superior.

Moreover, some typos and grammatical errors should be revised properly.

Author Response

Dear Reviewer:

Thank you very much for your careful review and constructive suggestions with regard to our manuscript “A virtual sensor for collision detection and distinction with conventional industrial robots”. Those comments are so valuable and helpful for authors to correct and improve the manuscript. We have studied comments carefully and tried our best to revise the manuscript and made corresponding changes in the manuscript. Appended to this letter is our point-by-point response to the comments that you raised.

Reviewer’s Comments

The authors presents a virtual sensor for collision detection and distinction in the human-robot interaction process.

- The idea is interesting and somewhat practical in the industrial

environment.

- The paper is well-organized and written.

- The result is supported by the experiments and analysed well.

- The limitations and future work is properly outlined.

1. Nevertheless, the authors are encouraged to compare their existing work with others, and also demonstrate why virtual sensor implementation is superior.

Response 1: Thanks for your valuable suggestion. We have add an experiment to compare the method proposed in our manuscript with the method proposed in references [16-17], highlighting the differences and advantages of the method proposed in this paper. Please see pages 18-19 of Section 4.3 of the revised manuscript.

2. Moreover, some typos and grammatical errors should be revised properly.

Response 2: Thank you very much for your careful review. We have carefully revised the full text of the manuscript and tried our best to correct typos and grammatical errors in the manuscript.

Round  2

Reviewer 2 Report

In this revised version, the authors have substantially made two modifications:

- a reference has been added and cited in the introduction

- the proposed method has been experimentally compared with a  previous one.

Almost nothing has been changed in the rest of the manuscript: English is still poor. Despite the authors’ efforts, several typos and errors are still present, and some further ones have been added in the new red parts.

All the part devoted to friction, which does not provide any original contribution but only the mere application of well known concepts, is still there as it was. It is quite obvious that an approach like the proposed one, based on constant thresholds, desperately relies on an accurate dynamic model that must necessarily include also an accurate friction model. So, where is the novelty?

I have some concerns about the introduced comparison with a previous method:

1) It is not correct to state that the comparison is made with respect to [16-17]. Only the method proposed in [16] (that is different from that in [17]) has been actually considered.

2) The comparison of the achieved results is quite superficial: statements like “The BPTOB observations are all near zero” are unacceptable from the scientific point of view, in my opinion. The sentence “the Low-pass filtering results is fluctuate relatively more” is wrong from the English point of view, and again too superficial for a scientific paper.

In my opinion, the manuscript still needs a deep revision before publication.

Author Response

Dear Reviewer:

Thank you very much for your careful review with regard to our manuscript “A virtual sensor for collision detection and distinction with conventional industrial robots”. Those comments are so valuable and helpful for authors to correct and improve the manuscript. We have carefully revised the full text of the manuscript. The first revision of the manuscript is marked in blue, and the part of this revision is marked in red. Appended to this letter is our point-by-point response to the comments that you raised.  

Comments and Suggestions for Authors

In this revised version, the authors have substantially made two modifications:

- a reference has been added and cited in the introduction

- the proposed method has been experimentally compared with a previous one.

Almost nothing has been changed in the rest of the manuscript: English is still poor. Despite the authors’ efforts, several typos and errors are still present, and some further ones have been added in the new red parts.

Thank you very much. We have carefully revised the sentence and grammar of the manuscript, including some unscientific and incorrect expressions in our manuscript. At the same time, in order to improve the quality of the manuscript, the manuscript has been further edited by MDPI.

All the part devoted to friction, which does not provide any original contribution but only the mere application of well known concepts, is still there as it was. It is quite obvious that an approach like the proposed one, based on constant thresholds, desperately relies on an accurate dynamic model that must necessarily include also an accurate friction model. So, where is the novelty?

Indeed, the main innovation of our manuscript is not in the friction model. As you said, the main purpose of the friction model in our manuscript is to improve the accuracy of the robot dynamics model, so as to compensate for the defects caused by the constant threshold. From the perspective of constant threshold, our manuscript is not as good as the literature [17], but our manuscript uses the momentum observation method to construct the filter, without the inverse dynamics calculation and the secondary derivation for calculating the acceleration. It is helpful for reducing the calculation error and noise, and enhancing real-time performance.

I have some concerns about the introduced comparison with a previous method:

1) It is not correct to state that the comparison is made with respect to [16-17]. Only the method proposed in [16] (that is different from that in [17]) has been actually considered.

Response 1: Thank you very much. We have corrected the error in the manuscript. Please refer to the first sentence of the last paragraph on page 14 of the manuscript.

2) The comparison of the achieved results is quite superficial: statements like “The BPTOB observations are all near zero” are unacceptable from the scientific point of view, in my opinion. The sentence “the Low-pass filtering results is fluctuate relatively more” is wrong from the English point of view, and again too superficial for a scientific paper.

Response 2: Thank you very much for your careful review. We have revised the unscientific statements in the manuscript. Please see the last paragraph on page 15.

In my opinion, the manuscript still needs a deep revision before publication.

Thank you very much for your careful review. We have carefully studied the comments of the experts. We have inspected the whole manuscript, including some unscientific and incorrect expressions in our manuscript. In order to improve the quality of the manuscript, the manuscript has been further edited by MDPI. We hope to meet the requirements of experts.
